# Machine Learning Techniques Applied to the Study of Drug Transporters

**DOI:** 10.3390/molecules28165936

**Published:** 2023-08-08

**Authors:** Xiaorui Kong, Kexin Lin, Gaolei Wu, Xufeng Tao, Xiaohan Zhai, Linlin Lv, Deshi Dong, Yanna Zhu, Shilei Yang

**Affiliations:** 1Department of Pharmacy, First Affiliated Hospital of Dalian Medical University, Dalian 116011, China; kxrxiaorui@163.com (X.K.); linkexin2022@163.com (K.L.); taoxufeng.2008@163.com (X.T.); hanhanjiayoudl@163.com (X.Z.); lvlinlinyu@163.com (L.L.); deshidong@163.com (D.D.); 2Department of Pharmacy, Dalian Women and Children’s Medical Group, Dalian 116024, China; wugao.lei@163.com

**Keywords:** machine learning, drug transporters, inhibiter, substrate

## Abstract

With the advancement of computer technology, machine learning-based artificial intelligence technology has been increasingly integrated and applied in the fields of medicine, biology, and pharmacy, thereby facilitating their development. Transporters have important roles in influencing drug resistance, drug–drug interactions, and tissue-specific drug targeting. The investigation of drug transporter substrates and inhibitors is a crucial aspect of pharmaceutical development. However, long duration and high expenses pose significant challenges in the investigation of drug transporters. In this review, we discuss the present situation and challenges encountered in applying machine learning techniques to investigate drug transporters. The transporters involved include ABC transporters (P-gp, BCRP, MRPs, and BSEP) and SLC transporters (OAT, OATP, OCT, MATE1,2-K, and NET). The aim is to offer a point of reference for and assistance with the progression of drug transporter research, as well as the advancement of more efficient computer technology. Machine learning methods are valuable and attractive for helping with the study of drug transporter substrates and inhibitors, but continuous efforts are still needed to develop more accurate and reliable predictive models and to apply them in the screening process of drug development to improve efficiency and success rates.

## 1. Introduction

Drug transporters are a group of transmembrane proteins that are widely distributed throughout the human body. They facilitate the movement of endogenous and exogenous substances into and out of biofilms, thereby influencing drug absorption, distribution, metabolism, excretion, and other pharmacokinetic processes. The investigation of transporters holds great importance in relation to pharmacokinetics, pharmacodynamics, drug–drug interactions (DDIs) and drug toxicity. Over 400 transporters have been identified in the human genome [1], primarily belonging to two superfamilies: the ATP-binding cassette (ABC) and the solute carrier (SLC) transporter. Over nearly two decades, various in vitro, in situ/ex vivo, and in vivo methods have been developed to study transporter function and drug–transporter interactions for the identification of their substrates or inhibitors. In vitro models comprise membrane-based and cell-based assays, whereas in vivo models encompass transporter gene knockout, natural mutant animal models, and anthropogenic animal models. In situ/in vitro models pertain to isolated and perfused organs or tissues, such as the liver, kidney, intestine, lung, and brain [2]. Although traditional research methods are constantly updated and improved, their experimental costs and time consumption remain significant obstacles in the research process, which is also a common challenge encountered during drug development. In addition, computational methods such as virtual screening (VS) and molecular docking are also employed in the study of drug transporters [3]. Molecular docking is a computational tool that enables the prediction of ligand conformation and binding affinity, as well as the identification of drug side effects and toxicity. Initially developed for investigating molecular recognition between small and large molecules, molecular docking has gained increasing popularity in supporting drug discovery programs. Its applications include but are not limited to hit identification and optimization, drug repositioning, a posteriori target identification (reverse screening), multi-target ligand design, and repositioning [4]. Common molecular docking platforms include DockThor-VS, Durrant Lab, iGEMDOCK [5], and AutoDock [6]. The proliferation of data and improvements in analysis techniques have given rise to ML-based prediction models, which present a valuable opportunity for investigating transporters [7]. These models can effectively address challenges in traditional research methods, providing advantages such as costs and time savings, as well as overall efficiency.

Currently, an increasing number of machine learning models are being developed in the field of transporter research. In this review, we focus on the application of machine learning in the study of transporters, with particular emphasis on recent advances in predicting transporter substrate/inhibitor interactions using machine learning models, in order to provide a reference for and help with the study of transporters.

## 2. Drug Transporters and Important Implications

Transporters are a ubiquitous class of proteins that are located on the cell membrane and facilitate transport functions throughout the human body (Figure 1). Until now, it has been generally assumed that multispecific drug transporters are derived from two transporter superfamilies: the ABC superfamily and the SLC superfamily. ABC transporters are a family of efflux transporters that transport drugs and endogenous substances by reversing the energy concentration gradient after ATP hydrolysis and are related to drug bioavailability, tumor multidrug resistance, and disease. Among the ABC family members, P-glycoprotein (P-gp), multidrug resistance-associated protein (MRP), and breast cancer resistance protein (BCRP) are considered to be important causes of multidrug resistance (MDR) of tumor cells [8] and therefore are the most studied subtypes related to drug transport. In addition, there are bile salt transporters (BSEP). Bile salt transporters that are not functioning properly or are expressed abnormally have been identified as significant factors contributing to various liver diseases, particularly those causing cholestasis [9]. Most SLC transporters are located on the cell membrane and rely on electrochemical and ion concentration gradients to transport substrates, regulate the exchange of soluble molecular substrates between the two sides of the lipid membrane, and maintain the stability of the intracellular environment. Over 400 transporters have been identified to date, displaying a wide range of substrates such as sugars, amino acids, vitamins, nucleotides, metals, inorganic ions, organic anions, oligopeptides, and drugs [10]. The SLC22 transporter family is among the most extensively researched SLC families in terms of drug handling [11], playing a central role in the transport of small molecule endogenous substances, drugs, and endotoxins across tissues and interfacial fluids. SLC transporters involved in drug transport are primarily composed of organic anion transporters (OATPs), organic anion transporters (OATs), organic cation transporters (OCTs), and oligopeptide transporters (PEPTs). Another SLC transporter, the multidrug and toxic compound efflux transporter (MATE), is an efflux transporter. Transporters expressed in the intestine, liver, and kidney play a critical role in the drug absorption, distribution, metabolism, and excretion (ADME) process. These transporters play a crucial role in regulating drug concentrations in both blood and tissues. Oral medication is primarily absorbed in the gastrointestinal tract, and its bioavailability is influenced by both uptake and efflux transporters present in this region. PEPT1, a transport protein expressed on the brush border membrane of the intestine, facilitates the absorption and transportation of peptide-like anticancer drugs within the gut. Linking the drug with a dipeptide can improve its bioavailability in the human body [12]. P-gp is the most extensively studied efflux transporter and plays a crucial role in limiting the bioavailability of numerous orally administered drugs [13]. MRP2 and BCRP are also expressed in the intestinal tract, with known substrates including statins, methotrexate, and other compounds. Transporters play a significant role in drug tissue distribution and elimination, ultimately influencing drug selectivity. Within the blood–brain barrier (BBB), various transport proteins, including P-gp, BCRP, and OCTs, play crucial roles in the distribution of neuroactive drugs. These transport proteins regulate the velocity and direction of drug transportation across the BBB. P-gp and BCRP can collaborate to facilitate the transportation of chemotherapy drugs [14].

SLC family members such as OCT1, OAT2, and NTCP are responsible for drug uptake into liver cells [15], whereas transport proteins involved in drug hepatobiliary efflux include P-gp, MRP2, BSEP, and BCRP [16,17]. There are many transporters (OCT, OAT, OATP, PEPTs, etc.) expressed on renal tubular epithelial cells that participate in proximal tubular secretion and reabsorption processes. These transporters play a crucial role in transferring drugs or their metabolites into urine for excretion [18]. In summary, alterations in transporter function can affect the ADME process and consequently drug efficacy, with transporters playing a crucial role in pharmacokinetics.

Transporters also play a crucial role in DDI by modulating the disposition of drugs within the body. DDI occurs when a drug influences the action of another drug by inhibiting or inducing one or more processes. Transporter-mediated DDIs, particularly those involving transporters expressed in the intestine, liver, kidney, and BBB, have garnered significant attention. DDI is likely to occur when the co-administered drug is a substrate, inhibitor, or inducer of the transporter protein. Through machine learning techniques, we can find the substrate or inhibitor of the transporter in a more efficient way, which can help us to better understand drug interactions during transporter studies, which can be important for both drug development and basic medical research.

## 3. Machine Learning

Artificial intelligence (AI) is widely utilized, leveraging computational power to emulate human cognitive processes. Machine learning, a pivotal component of artificial intelligence, can be traced back to 1943 [19]. The term refers to the capability of software to accomplish a task by means of learning from data and has been widely employed in various domains, such as data integration and analysis [20,21]. Machine learning possesses the ability to identify complex patterns from vast and complex molecular descriptor datasets, making it particularly suitable for predicting transporter substrates and inhibitors. Depending on the type of data, such as whether sample labels are available, machine learning algorithms can be classified as supervised, semi-supervised, and unsupervised learning [22]. When machine learning is used to predict transporter substrates or inhibitors, it is often done through supervised learning, where models are built using labeled training data. Model building can include options such as decision trees, random forests, neural networks, support vector machines, logistic regression, k-nearest neighbors, and more. Each model has its own characteristics and suitable environments.

### 3.1. Decision Trees and Random Forests

Tree-based algorithms are very popular in machine learning and are a method of classification and regression using decision trees [23]. Decision tree learning is a supervised learning technique based on the concept of recursive classification. In this method, classification models are represented as tree structures that start at a decision point and use a feature that can split the data. Each split is connected to a new decision point that contains more features to further separate the data. In addition to simple decision trees, there are newer ensemble methods, such as random forest (RF) and gradient boosting trees (XGBoost). Random forest builds multiple decision trees and combine their prediction results to improve prediction accuracy and prevent overfitting. Each decision tree is created using a subsample of features, not each feature [24].

### 3.2. Neural Network

Artificial neural network (ANN), deep neural network (DNN), and deep learning (DL) are also common algorithms in the field of machine learning. The concept is grounded in the architecture of the human brain and can be effectively applied to both regression and classification problems. An Ann model consists of units that combine multiple inputs and produce a single output, including an input layer, an output layer, and a hidden layer between them, each consisting of multiple neurons in parallel. The existence of hidden layers enables the categories in the input signal that are not linearly separable to be distinguished. The nonlinear activation function modifies the signal of the input node and outputs it to the next node, each output node corresponds to the task to be predicted, and finally, the complex information is classified [25]. DNNs are artificial neural networks with multiple hidden layers, which are considered deep learning algorithms with more complex networks and data volumes, so the problem of overfitting needs to be considered. There are several well-known variants of deep learning, such as convolutional neural networks, recurrent neural networks (RNNs), and so on [26].

### 3.3. Support Vector Machine

Support vector machine (SVM) is a kind of machine learning with maximization (support) of separating the margin (vector), which is a classical nonlinear classification and regression modeling algorithm. The separation hyperplane is constructed in space, the distance between the separation hyperplane and the nearest expression vector is defined as the edge of the hyperplane, and the classification ability is maximized by selecting the maximum edge to separate the hyperplane. Constructing the optimal hyperplane requires support vectors and some training data [27,28]. Achieving the optimal separation requires the application of kernel functions, which can add additional dimensions, and the data become better separated in the high-order space, which is also an advantage of SVM.

### 3.4. Naïve Bayes

Naïve Bayes (NB) uses Bayes’ theorem to classify data under the assumption that each feature of a sample is uncorrelated (strongly independent) with other features. Compared with other machine learning algorithms, the Bayesian algorithm is a faster and simpler algorithm, only needs to consider each predictor variable in each class separately [29], and has relatively low accuracy, so it can perform better on less complex data.

### 3.5. k-Nearest Neighbor Algorithm

The k-nearest neighbor algorithm (k-NN) is a machine learning algorithm mainly used for classification and is widely used due to its simple and easy-to-understand design [30]. It classifies unlabeled data by assigning them to the most similar labeled category. The k-nearest training data (neighbors) are considered, and the final classification is determined and checked according to the majority voting rule [31]. Factors such as the k value, distance calculation, and appropriate predictor variable selection can all have a significant impact on model performance [32].

The general procedure for identifying new substrates/inhibitors of drug transporters through machine learning techniques is outlined as follows: (1) A database is built of known compounds as substrates/inhibitors as a dataset; (2) the chemical information of the compound is analyzed, extracted, and converted into a form that can be recognized by the algorithm; and (3) the constructed dataset is split into a training set and a validation set. Machine learning methods are employed to learn from the training set and develop the model, whereas the validation set is used to test and enhance the newly created model. (4) The unknown compounds are predicted and verified.

## 4. Application of Machine Learning Methods in the Investigation of Drug Transporters

### 4.1. ABC Transporters

Many human ABC proteins are efflux transporters, including P-gp (ABCB1), MRPs (ABCC), and BCRP (ABCG2), and function as efflux pumps that actively extrude compounds such as drugs from the cell. The classical ABC transporter is structurally composed of four structural domains, two transmembrane domains (TMDs), and two cytoplasmic nucleotide-binding domains (NBDs) [33]. Transporter proteins associated with MDR belong to the ABC transporter superfamily, which is one of the major barriers to cancer therapy and affects drug accumulation in cancer cells [34]. Among these transporters, P-gp is considered to be the major contributor to cellular multidrug resistance. The tissue distribution and cellular localization of transporters influence drug efficacy and toxicity. Therefore, it is essential to study the efflux transport of drugs and identify the substrates of efflux transporters. Additionally, exploring efflux transporter inhibitors represents a promising research direction for addressing drug resistance. The machine learning methods used by the researchers are listed in Table 1.

#### 4.1.1. P-gp

The ABCB1 transporter, also known as P-gp, belongs to the ABCB subfamily. It was first identified in Chinese hamster ovary cells in 1976 [55]. With the introduction of the concept of the ABC transporter family [56], the research on P-gp gradually increased. There are two genes that encode P-gp in humans: MDR1 and MDR1A/1B P-gp, which are mainly distributed in human small intestine, colon, liver, kidney, brain, and other tissues and organs, as well as barrier tissues such as the blood–brain barrier, blood–testis barrier, and placental barrier. They are also expressed in the lung, heart, and spleen [57]. P-gp functions as an efflux transporter for endogenous substances, exogenous substances (drugs and their metabolites), and toxins out of cells. Therefore, in normal tissues, P-gp-mediated efflux transport helps to reduce toxicity and protect cells, but at the same time, it limits the absorption of drugs and reduces the bioavailability [58]. P-gp is highly expressed on the membrane of many tumor cells, which is directly related to the multidrug resistance of tumor cells. Not only anticancer drugs but also HIV protease inhibitors and immunosuppressants are the substrates of P-gp [59]. Therefore, drugs that inhibit P-gp are anticipated to elevate the intracellular concentration of chemotherapeutic agents and enhance their sensitivity.

P-gp is the earliest discovered transporter [60] and has been studied for about 30 years. Therefore, there is a large amount of data accumulation on P-gp transporters, and most machine learning methods in the early stage are carried out around P-gp. With the development of computer technology, machine learning prediction models of P-gp have been constantly improved.

In 2019, Kadioglu et al. [35] established a prediction platform for P-gp modulators using machine learning methods (including k-NN, neural networks, RF, and SVM). They used defined chemical descriptors to predict whether test compounds can act as substrates or inhibitors of P-gp. It is noteworthy that they also validated the results using molecular docking in terms of binding energy and docking poses. The RF classification algorithm performed better than other algorithms in feature selection. In 2020, Esposito et al. [36] combined machine learning with MD simulations using the MDFP/ML approach, using molecular dynamics fingerprints (MDFPs) as orthogonal descriptors to distinguish and predict substrates and non-substrates of P-gp. The study used four different ML methods, namely, RF, GTB, SVM, and meta-learner. When the model was validated with an external validation set, it was found that only models trained on MDFPs or attribute-based descriptors could be applied to chemical space areas not covered by the training set. Despite P-gp being a well-known entity for over three decades, the lack of improved selective inhibitors targeting this protein can be attributed to its specificity and unknown structural characteristics.

#### 4.1.2. BCRP

BCRP, a member of the G subfamily of the ABC family, was first identified in the multidrug resistant human breast cancer cell line MCF-7/AdrVp [61]. BCRP is widely expressed and distributed in several normal tissues, such as the small intestine, liver, brain endothelium, and placenta [62]. It can confer resistance by pumping chemotherapy drugs out of cells. In the past decade, the research of machine learning in BCRP has developed rapidly.

Hazai et al. [44] developed an SVM prediction model of BCRP substrate based on the known substrates and non-substrates of BCRP in 2013. For model verification, a training set/testing set machine ratio of 0.75/0.25 was chosen, and the overall prediction accuracy for the independent external validation dataset was 40%. Moreover, the prediction accuracy for the wild-type BCRP substrate was higher than that of the non-substrate, with a rate of 76%. The 3D structure of the substrate was found to be a possible determinant of the BCRP–substrate interaction by the molecular descriptors it used. In 2014, Ding et al. [45] developed an accurate, fast, and robust pharmacophore ensemble/support vector machine (PhE/SVM) model to predict the BCRP inhibition of structurally diverse molecules. Due to the confounding nature of BCRP, this method does not produce significant bias when applied to various structurally diverse inhibitors. In 2016, Montanari et al. [41] integrated data using KNIME workflows to build a multi-label classification model of BCRP/P-gp inhibitory activity using a machine learning approach. Key molecular features affecting transporter selectivity were retrieved by comparing various multi-label learning algorithms. The KNIME workflow is an effective solution for merging data from multiple sources and constructing multi-label datasets that are tailored for BCRP and/or P-gp. Using the dataset created through the KNIME workflow, it was possible to distinguish between selective BCRP inhibitors and selective P-gp inhibitors by examining only two features: the count of hydrophobic and aromatic atoms, and the shared characteristics between dual and selective inhibitors. In 2017, Gantner et al. [42] were the first to combine computer predictions of BCRP with experimental validation to develop nonlinear computer models of BCRP substrates. The J48 decision tree induction algorithm implemented by the C4.5 decision tree algorithm in WEKA3.651 is used to obtain the corresponding nonlinear classification model, and the genetic algorithm (GA) is used to select the best descriptor. The selected non-substrate compounds were experimentally validated using a stereovalgus rat intestinal sac model, which demonstrated the predictive power of the model. The rfSA technique is a feature selection approach that uses both the simulated annealing (SA) algorithm and RF to eliminate redundant and irrelevant features. In 2020, Jiang et al. [43] used XGBoost and DNN methods for the prediction of BCRP inhibitors for the first time and obtained good prediction results. A diverse set of 1098 BCRP inhibitors and 1701 non-inhibitors was compiled as a dataset, and the molecular descriptors linked to BCRP inhibition were explored. It was found that one of the characteristics of BCRP inhibitors was high hydrophobicity and aromatic properties. Seven ML methods (DNN, SGB, XGBoost, NB, weighted k-NN, RLR, and SVM) were used to develop the classification model. The Bayesian optimization algorithm was used to optimize the hyperparameters. The results showed that the SVM, XGBoost, and DNN methods were superior to other methods, and SVM had the best prediction ability. Analysis of the misclassified compounds revealed that most of them had complex structures and may not be able to be accurately characterized by traditional descriptors. In 2021, Ganguly et al. [40] used a Bayesian machine learning model to predict the metabolites most likely to be BCRP or P-gp substrates in CSF and plasma of dKO rats, demonstrating that CSF may be a better substrate for identifying endogenous substrates of BCRP and P-gp.

BCRP has been shown to have a role in the permeability of the blood–brain barrier, resulting in the failure of most CNS-acting drugs in clinical trials [63]. In 2014, Garg et al. [46] developed a machine learning model to evaluate the effect of BCRP on BBB, an artificial neural network model to predict the BBB permeability of molecules, and an SVM model to predict the substrate of BCRP. Through molecular docking analysis, 11 molecules were identified as meeting the criteria for BBB penetration. Additionally, these compounds were predicted to be substrates of BCRP in BBB permeability.

#### 4.1.3. MRPs

MRPs are active transporters of the ABC family and are widely distributed in the lung, kidney, brain, and other organs. MRPs contain many isoforms, among which MRP1, MRP2, and MRP4 are highly expressed in tumor cells and mediate the efflux of a variety of anti-tumor drugs, leading to the occurrence of multidrug resistance. Therefore, the study of MRPs is highly significant in combating multidrug resistance in tumors. Recently developed machine learning methods for predicting substrates or inhibitors of MRPs have demonstrated remarkable accuracy.

In 2017, Lingineni et al. [48] established a SVM model for MRP1 substrate classification based on previous studies [46], and the accuracy of the best MRP1 substrate model in the training set, test set, and external validation set was 87.39, 93.54, and 80%, respectively. The BBB permeability artificial neural network model and molecular docking analysis demonstrated that MRP1 plays an important role in the transport of substances in the BBB.

Kharangarh et al. [47] used k-NN, RF, SVM, and other machine learning methods to train the classification model of MRP2 inhibitors and non-inhibitors using compounds from the Metrabase database and obtained different descriptors through four methods: variance threshold, SelectKBest, RF, and REF. The k-NN, RF, and SVM methods were used to train the machine learning model. The five-fold cross-validation and analysis of relevant parameters showed that the SVM model constructed by the features selected by the RFE method performed well, and the key descriptors for developing MRP2 inhibitor models were obtained by RStudio analysis, which could determine the inhibitory properties of the MRP2 protein in the early stages of drug discovery. The prediction of MRPs substrates can reduce the failure rate of preclinical drug studies, and the prediction of inhibitors can help the study of MDR, both of which can be used in the early stages of drug discovery.

#### 4.1.4. BSEP

BSEP is a kind of ABC transporter encoded by the ABCB11 gene. It is located in the duct membrane of hepatocytes and is responsible for transporting bile acids and bile salts from hepatocytes to bile tubules [64]. Inhibition of BSEP can cause the toxic accumulation of bile salts in cells, triggering cholestatic liver injury and ultimately leading to the premature termination of preclinical development and clinical trials of drug candidates. Machine learning prediction of BSEP has also been studied in recent years. In 2021, McLoughlin et al. [49] developed a model for predicting and classifying BSEP inhibitors. They utilized the Automated Data-Driven Modeling Pipeline (AMPL) to train and assess over 15,500 classification and regression models. The optimal combination of model types, dataset segmentation strategies, chemical characterization methods, and model parameters is determined by testing various configurations using the AMPL’s hyperparameter search function. The best performing model for this purpose was finally found to be the RF model, which included MOE descriptor features.

### 4.2. SLC Transporters

More than 400 transporters have been identified, and the SLC22 transporter family is one of the best studied SLC families for drug handling, with a central role for small molecule endogenous substances, drugs, and endotoxins that move between tissues and interfacial fluids. The kidney expresses high levels of OCT2 and OAT1, which are crucial for the renal uptake of clinical drugs and endogenous substances. OAT1, OAT3, OCT1, and OCT2 are widely studied drug transporters. OAT2 is mainly expressed in hepatocytes and involved in the transport of small molecule anion drugs to hepatocytes. OAT1 and OAT3 are mainly expressed in renal cells and regulate the transport of organic anions from the blood into proximal tubular cells. OCT1 is mainly expressed in the hepatic sinusoidal membrane and mediates the transport of drugs and endogenous substances. OCT2 and OCT3 are involved in the renal and biliary excretion of cationic drugs, respectively. In 2016, Ose et al. [39] developed a model for predicting drug transporter substrates based on the SVM method and established a database of seven classes of transporter substrates (OATP1B1/1B3, OAT1/3, OCT1/2, MRP2/3/4, MDR1, BCRP, and MATE1/2-K). Physicochemical parameters were used as the basic descriptors. This model has the potential to accurately predict transporter substrates without the need for in vitro transport assays. In 2017, Shaikh [37] performed multi-transporter modeling and developed substrate prediction models for transporters using quantitative structure–activity relationship (QSAR) and protein stoichiometry (PCM) methods. After evaluating the established models, the top-performing model was merged with other models to create a heterogeneous integrated model for each transporter. This analysis involved 6 efflux transporters, 7 uptake transporters, and 4575 substrate/non-substrate data. In 2020, Nigam et al. [50] combined machine learning, chemoinformatics, and multi-specific drug transporter knockout metabolomics to analyze the unique metabolites accumulated in the plasma of OAT1 and OAT3 knockout mice and define the molecular properties of endogenous ligands. Finally, seven key molecular descriptors were obtained. The RF classification model based on the metabolite dataset correctly classified ≥ 75% of the drugs known to interact with OAT1/3. This helps with the physiological role of drug transporters, metabolite-based drug design, and analysis of drug–metabolite interactions. This cheminomics–machine learning approach was subsequently used to analyze OATP and OAT-transported drugs by Nigam et al. [51]. The results showed that liver OATPs preferred highly hydrophobic, complex, and more ring-like drugs as substrates, whereas kidney OATs preferred more polar drugs. This provides a molecular basis for tissue-specific targeting strategies, drug interactions, and drug delivery to minimize toxicity in liver and kidney diseases. In 2021, Jensen et al. [54] used machine learning methods to predict substrates for OCT1. A database containing more than 1000 substances was predicted by virtual screening, and 19 substances were tested in vitro. This study demonstrates that machine learning methods can accurately predict substrates of OCT1, even in the absence of its crystal structure.

The norepinephrine transporter (NET/SLC6A2) is also a member of the SLC family and is more well studied than other SLCS. NETs can regulate NE-mediated physiological effects by terminating noradrenergic signaling by uptake of norepinephrine into presynaptic terminals. In 2023, Bongers et al. [65] developed a technique for the identification of NET inhibitors combined with machine learning methods using RF, GBt, and PLS algorithms to build a model combined with virtual screening and experimental validation and finally identified five novel NET inhibitors. This method incorporates the chemical space of the ligand and utilizes a similarity-based network to select related proteins for the modeling of NETs.

Obtaining data is sometimes limited by privacy issues. Collecting and sorting data also requires a lot of time and effort. After obtaining data, it is also crucial to select chemical descriptors that can establish high-quality models for different transporters. To make it possible to build predictive models in a quick and easy way, Smajic et al. [38] worked with Jupyter Notebooks in 2022 to create a framework that can create or retrain ML models in a semi-automatic manner. Classification models of six transporters (BCRP, BSEP, OATP1B1, OATP1B3, MRP3, and P-gp) were allowed to be generated, and the created models could be updated and shared using Jupyter Notebooks. This is a valuable tool for predicting new data on ABC and SLC transporters. Table 2 shows the sources of the datasets used by the researchers to build their machine learning models.

## 5. Conclusions and Future Prospects

Advances in information technology have created new methods for advances in many fields, including pharmaceutical research. Cost and efficiency are also a major challenge in the drug discovery process. Drug resistance is one of the main reasons for the failure of chemotherapeutic drugs, so predicting whether a chemotherapeutic drug is a substrate of a transporter or not is essential; the alteration of drug efficacy and toxicity caused by DDI can be predicted by the identification and prediction of transporter substrates and inhibitors. Nowadays, computerized classification models of various transporter substrates and inhibitors have been established to save experimental resources. They have been instrumental in overcoming drug resistance, DDI discovery, and drug targeting.

The quality of the data has a significant impact on the performance of the model. In addition to the machine learning models that have been built so far in the initial database establishment stage, as well as obtaining data from public databases, we need to manually collect and collate data from different studies or use unpublished data from our own laboratory. This may affect the reliability of the prediction results. Therefore, the selection of descriptors and validation of the model is also an important step to ensure the accuracy of the prediction results.

In this paper, we have reviewed the machine learning technique-based approach to study different transporters. The use of a single source of data and construction of models to understand the role of drugs and transporter proteins is not sufficiently accurate. Classification efficiency and higher predictive accuracy of machine learning models depend on comprehensive and reliable data and trade-offs between individual machine learning approaches. In addition, further validation is needed for transporter substrates and inhibitors predicted through machine learning. In general, machine learning provides a highly useful tool for studying transporters, improving research efficiency, and allowing us to focus on compounds with higher potential.

## Figures and Tables

**Figure 1 molecules-28-05936-f001:**
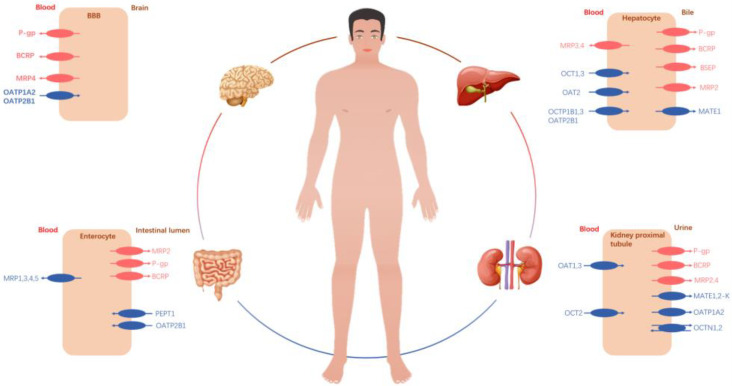
Expression of ABC and SLC transporters with major roles in drug efficacy or toxicity in human intestinal epithelia, hepatocytes, kidney proximal tubule epithelia, brain capillary endothelial cells, and choroid plexus epithelial cells.

**Table 1 molecules-28-05936-t001:** Application of machine learning methods in the investigation of drug transporters.

Transporter		ML Methods	References
ABC	P-gp	RF	[35,36,37,38]
		NN	[35]
		SVM	[35,36,37,38,39]
		k-NN	[35,37,38]
		Bayes	[37,40]
		Logistic regression (LR)	[37,38]
		GTB	[36]
	BCRP	RF	[37,38,41,42]
		Bayes	[37,40]
		DNN	[43]
		SVM	[37,38,39,41,44,45,46]
		k-NN	[37,38]
		XGBoost	[43]
		LR	[37,38,41]
	MRPs	RF	[37,38,47]
		SVM	[37,38,39,47,48]
		Bayes	[37]
		k-NN	[37,38,47]
		LR	[37,38]
	BSEP	RF	[38,49]
		SVM, LR, k-NN	[38]
SLC	OAT	RF	[50,51]
		SVM	[39]
		SNN, NB, k-NN, LR	[51]
	OATP	RF	[52,53]
		k-NN, LR	[37,38,51,53]
		XGBoost, DL	[53]
		SVM	[37,38,39]
		SNN	[51]
		Bayes	[37,51,53]
	OCT	RF, Bayes, k-NN, LR	[37]
		SVM	[37,39]
	MATE1,2-K	SVM	[39]
	NET	RF	[54]

**Table 2 molecules-28-05936-t002:** The source of the dataset.

Transport Protein	Data Sources	References
P-gp	Literature	[35]
P-gp	In-house dataset; ChEMBL [66]	[36]
P-gp, BCRP, MRPs, OATP, OCT	Metrabase [67] (http://www-metrabase.ch.cam.ac.uk, accessed on 25 July 2023); literature	[37]
P-gp, BCRP, MRPs, BSEP, OATP	LiverTox [68]; ChEMBL and PubChem [69]	[38]
P-gp, BCRP, MRPs, OAT, OATP, OCT, MATE1,2-K	Text-mining technique [70]; TP search (http://togodb.dbcls.jp/tpsearch, accessed on 25 July 2023); DIDB (http://www.druginteractioninfo.org/, accessed on 25 July 2023); PharmGKB (www.pharmgkb.org); TransPortal (http://dbts.ucsf.edu/fdatransportal, accessed on 25 July 2023); PubChem	[39]
P-gp, BCRP	Literature	[40]
BCRP	Literature; the Open PHACTS Discovery Platform	[41]
BCRP	Literature	[42,43,45,46]
BCRP	Literature; University of Washington Metabolism & Transport Drug Interaction Database (http://www.druginteractioninfo.org/, accessed on 25 July 2023); PubChem Database (http://pubchem.ncbi.nlm.nih.gov, accessed on 25 July 2023)	[44]
MRPs	Literature; Metrabase	[47]
MRPs	Literature; PubMed; TP search	[48]
BSEP	A proprietary BSEP assay dataset; published dataset [71]	[49]
BSEP	Training dataset	[72]
OAT	Literature; training dataset	[50]
OAT, OATP	PubChem	[51]
OATP	ChEMBL, UCSF-FDA TransPortal, DrugBank, Metrabase, IUPHAR	[52]
OATP	ChEMBL	[53]
OCT	Literature	[73]
OCT1	Training dataset	[74]
NET	ZINC database [75]; PubChem; literature	[54]
SGLT1	ChEMBL; the Spectrum Collection compound library	[76]

## Data Availability

No experimental data are available for this review.

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
