# Peer review of "Machine Learning Techniques Applied to the Study of Drug Transporters"

_molecules, 2023, doi:10.3390/molecules28165936_

Round 1
Reviewer 1 Report
It is a good idea to publish a review about the modeling of transport & efflux systems, which are paramount players in pharmacokinetics. However, the 14 manuscript pages expedite the issue a little bit to quickly in my opinion. This is more of a bibliographic survey, not so much a review. For example, it would be - from the reader's point of view - to clearly separate publications including experimental validation of their predictions from those just reporting a model with nice statistics and the frivolous conclusion "This COULD be helpful for drug design" (frankly, I would not even bother to review those, anyway). Which tools are commercial, and which tools are freely accessible? Would be nice to have a table with the links towards those websites providing efflux/transport -related prediction. Writing a review is not simply listing what was done in literature - it is also providing a critical assessment of the done work and, foremost, the quality of the DATA used to build the models. Or, we are not told much about the data sets (maybe links from where to download these would be a great plus). At the end of the day - what models would YOU use? Have you had the occasion to independently validate or disprove some of their predictions?
English can always be improved, but the article seemed quite readable to me...
Reviewer 2 Report
The manuscript content is interesting. However, some points should be addressed:
a) The review should be original, please highlight the content which can be different to other documents in the field.
b) From my perespective, references in table 1 are the most attractive content. However, just near of 15 articles are included, while more than one hundred could be found in google scholar in this topic.
c) Discussion and Conclusion shoul be focused in the explored field. They seem to be general and not applied to transporters.
Minor changes
Round 2
Reviewer 1 Report
The authors have done some improvement and added an important table with links to data - they could have done more, but after all they are the authors, up to them to design the article they way they like.
Fine enough - nothing that would make William Shakespeare raise from his grave in shock
Reviewer 2 Report
The manuscript is improved. Please clarify if literature meaning in Table 2 is "journals and books"?
Minor details